# *Relish* as a Candidate Marker for Transgenerational Immune Priming in a Dampwood Termite (Blattodae: Archeotermopsidae)

**DOI:** 10.3390/insects11030149

**Published:** 2020-02-27

**Authors:** Erin L. Cole, Jessica S. Empringham, Colette Biro, Graham J. Thompson, Rebeca B. Rosengaus

**Affiliations:** 1Department of Marine and Environmental Sciences, Northeastern University, 134 Mugar Life Sciences Building, 360 Huntington Avenue, Boston, MA 02115, USA; cole.eri@husky.neu.edu (E.L.C.); biro.c@husky.neu.edu (C.B.); 2Department of Biology, Western University, 1151 Richmond St. London, ON N6A 5B7, Canada; jempringham2020@meds.uwo.ca (J.S.E.); graham.thompson@uwo.ca (G.J.T.)

**Keywords:** social insect, immunity, embryonic defenses, transgenerational immunity, IMD pathway, gene expression, hemimetabolous, parental effects

## Abstract

Natural selection should favor the transfer of immune competence from one generation to the next in a context-dependent manner. Transgenerational immune priming (TGIP) is expected to evolve when species exploit pathogen-rich environments and exhibit extended overlap of parent–offspring generations. Dampwood termites are hemimetabolous, eusocial insects (Blattodea: Archeotermopsidae) that possess both of these traits. We predict that offspring of pathogen-exposed queens of *Zootermopsis angusticollis* will show evidence of a primed immune system relative to the offspring of unexposed controls. We found that *Relish* transcripts, one of two immune marker loci tested, were enhanced in two-day-old embryos when laid by *Serratia*-injected queens. These data implicate the immune deficiency (IMD) signaling pathway in TGIP. Although an independent antibacterial assay revealed that embryos do express antibacterial properties, these do not vary as a function of parental treatment. Taken together, *Z. angusticollis* shows transcriptional but not translational evidence for TGIP. This apparent incongruence between the transcriptional and antimicrobial response from termites suggests that effectors are either absent in two-day-old embryos or their activity is too subtle to detect with our antibacterial assay. In total, we provide the first suggestive evidence of transgenerational immune priming in a termite.

## 1. Introduction

Mendelian genetics have dominated our perception of the laws of inheritance: the progeny’s phenotype is shaped by instructions inherited from the parental DNA. However, within the past ~20 years, a fundamental shift in our understanding of heritable phenotypes has taken place—namely, that some environmentally acquired information may also be transferred between generations, from parent to offspring and even across subsequent generations. Such context-dependent inheritance does not involve nucleotide changes in the genetic code [1,2,3]. Instead, the parental transfer of nutrients, hormones, cytoplasmic factors, small RNAs and other information-bearing molecules, as well as epigenetic tags that can modulate offspring gene expression [4,5,6], results in changes to offspring phenotype that anticipate environmental challenges and their demands. For example, differences in parental environments stemming from altered photoperiods [7], resource availability [8], temperatures [3], social environments [9], predation pressure [10], and parasite load [4] can potentially influence offspring phenotype in a manner that is independent of offspring genotype. Through parental effects (both maternal and paternal across one generation [11]) or transgenerational effects (across multiple generations [1,2,3,12,13]), parents can convey environmentally acquired qualities to their offspring, thereby anticipating their offspring’s fit to their environment [2,12,13]. 

One example of potentially adaptive nongenetic inheritance is transgenerational immune priming (TGIP) whereby parents infected by a pathogen can immunologically prime their offspring against future infection [4]. TGIP is most likely to evolve when the pathogen is predictable in time and space (i.e., likely to persist into the next generation [5,14]) and when hosts exhibit either limited dispersal and/or overlapping generations [4,15]. If so, we expect TGIP to manifest via parental transfer of prefabricated immune proteins [16], immune-related transcripts [17], immune-elicitors [18,19], altered epigenetic markers [20,21], or combinations thereof, ultimately rendering progeny less susceptible to disease [4,5]. For example, immune-primed mothers of *Manduca sexta* (Lepidoptera) can translocate pathogen-associated molecular patterns (PAMPs; molecular signatures unique to invading microbes) to their eggs [18,21] which, upon hatching, produce larvae with enhanced immune-related phenoloxidase expression and improved bacterial clearing capabilities [22,23]. Similarly, offspring of *Tenebrio castaneum* (Coleoptera) show up-regulated antimicrobial peptides following maternal exposure to bacteria [24]. These mothers effectively prime their offspring by translocating bacteria from their gut to their eggs during vitellogenesis [18,19]. Fathers, too, can influence the offspring’s immune function [4]. For example, Eggert, Kurtz, and Diddens-de Buhr [25] demonstrated that bacterially exposed male red flour beetles sired offspring with enhanced phenoloxidase levels and higher survival against an immune challenge. 

Evidence for TGIP in insects predominantly comes from solitary, holometabolous insect species [4,5,18,19,26,27,28]. However, social insects (mostly in the orders Hymenoptera and Blattodea [29]) live in densely populated groups of close relatives, rendering them highly vulnerable to the spread of contagion [30,31]. Although TGIP is present in several lineages of the social Hymenoptera, including bumble bees [14,32,33], honey bees [34,35], and two species of ants [36,37], no study has tested for evidence of TGIP in termites (Blattodea, Isoptera), a hemimetabolous eusocial taxon. Not only do termites exhibit low dispersal and overlapping generations, but dampwood termites in particular live under high microbial loads [38]. We therefore consider dampwood termites a good candidate taxon to study TGIP.

If TGIP evolved in dampwood termites, we predict its expression to be most acute during the early stages of colony foundation. Incipient colonies are most vulnerable to infection compared to other colony stages [39,40,41,42,43]. Given their lack of workers to assist with nest hygiene (i.e., allogrooming, deposition of antimicrobial compounds in nest material [44]) and social immunity [45,46], the health and well-being of the first-brood are completely dependent on the care and contributions made by parents, as well as the progeny’s own immune systems. Here, we experimentally tested for evidence of transgenerational immunity in termite embryos. We focused on measuring the effect that pathogenic stress experienced by one generation had on the immune competence of the next generation. First, we compared embryonic transcript abundance levels of *immune deficiency* (*IMD*) and *Relish,* which are two major components of the immune deficiency pathway [47]. Recognition of Gram-negative-associated peptidoglycans by a transmembrane peptidoglycan recognition protein results in the formation of a complex that includes the IMD protein. The recruitment of this complex results in the initiation of an enzymatic cascade leading to the activation of *Relish*, a factor that promotes the transcription of various downstream antimicrobial peptides known to be active against Gram-negative bacteria [47]. Hence, these two loci are required for both the initiation of the IMD immune response and its antibacterial action. Second, we assessed embryonic antibiotic activity when parents were healthy, immune elicited, or immune challenged. 

## 2. Materials and Methods 

### 2.1. Colony Collection and Establishment of Incipient Termite Colonies 

We collected 17 mature colonies of *Z. angusticollis* from the East Bay Regional Park District in Oakland, CA (USDA Permit P526P-17-03814). We maintained these field-collected colonies in the lab within Rubbermaid containers (25 °C and 60% RH). Between May and October, these colonies produced alates (winged reproductives). We removed heavily sclerotized alates from their natal colony and proceeded to sex, de-wing, weigh, and treat individual termites according to the scheme described in Figure 1 (sample sizes in Table 1). Following treatment, we established pairs of males and females (307 nestmates, 316 non-nestmates) inside Petri dishes (60 × 15 mm) containing moist filter paper (Whatman #1) and ~2.5 mg of birch wood [48]. We then stacked all pairs inside covered plastic bins lined with wet paper to prevent desiccation (~90% RH).

### 2.2. Parental Treatments, Microinjections, and Embryo Collection 

*Serratia marcescens* is a Gram-negative, ecologically relevant, facultative pathogenic bacterium that is commonly associated with termites [49,50]. We provide details on the preparation of *S. marcescens* cultures in Appendix A. We haphazardly assigned de-winged kings and queens to one of the following four treatments: untreated (naïve), a 1 µL injection of sterile Burnes Tracey Saline (BTS; control), 1 µL of a 10^7^ CFU/mL heat-killed suspension of *Serratia* (effective dose of immune-elicitors of ~10,000 colony-forming units (CFUs); HK-*Sm*), or 1 µL of a live 2 × 10^5^ CFUs/mL *S. marcescens* suspension (translating into a known sublethal dose of ~200 CFUs/insect; Live-*Sm* [42]). To visually confirm that our treatments were successfully injected into the termites’ hemocoel, we added green food coloring to both control and experimental treatments (McCormick^®^; 5 mL per L of BTS). To facilitate injections and minimize the risk of septic injury, we first cold immobilized (5 °C) alates individually and sterilized their cuticles by swabbing their sternites with 70% ethanol, as in Cole et al. [42]. We injected termites with a picospritzer III (Parker Hannifin, Cleveland, OH, USA) fitted with a pulled borosilicate capillary tube (2 µm diameter) into the intersegmental membranes of the third to fifth sternites, and ultimately into the hemocoel. 

Following treatment, we paired kings and queens as described above. We inspected the incipient colonies for oviposition every second day. Following the onset of oviposition, we removed two-day-old embryos and immediately flash-froze them in liquid nitrogen. We then deep-froze embryos at −80 °C until we used them in three assays: transcript abundance analyses of embryos from maternal effects only; quantification of embryonic protein content from the maternal, paternal, or combined effects; and in vitro antimicrobial assay of embryos sired by either treated queens, treated kings, or treated queens and kings (Figure 1). Sample sizes for each assay are provided in Table 1. 

### 2.3. Transcript Analyses

To measure immune gene transcripts in embryos from treated mothers, we performed a digital drop quantitative PCR (ddPCR) experiment. ddPCR is more precise and sensitive than real-time PCR [51,52] and should be able to detect subtle differences in the gene expression of termite embryos. The ddPCR assay allowed us to quantify the relative transcript abundance of two effector proteins (*Relish* and *IMD*) that are integral to the immune deficiency (IMD) immune pathway as a function of maternal treatment. This pathway is a major arm of an insect’s humoral immune system, and its induction results in the transcription and translation of antimicrobial peptides which are active against Gram-negative pathogens, such as *Serratia* and *Arthrobacter* [47,53]. Genomic evidence indicates that *Zootermopsis nevadensis* (a congener of *Z. angusticollis*) possesses a functional IMD pathway, along with several Gram-negative-binding proteins [54]. We predicted, therefore, that *Z. angusticollis* embryos sired by challenged (Live-*Sm*) queens would show higher levels of both *Relish* and *IMD.*

We extracted RNA from embryos (*n* = 5 embryos per sample; Figure 1B). For each pooled embryo sample, we manually disrupted frozen tissue in an RNeasy Micro Kit (Qiagen, Hilden, Germany) lysis buffer (RLT, 350 µL) using disposable pestles from Sigma (Sigma-Aldrich, St. Louis, MO, USA) attached to a hand-held microtube homogenizer (Thomas Scientific, Swedesboto, NJ, USA). We captured the total RNA from the homogenate using an RNeasy Micro Kit in the manner prescribed for animal tissues. We eluted the final extracted RNA in a volume of 19 µL of RNAse-free water. Immediately following RNA extraction, we synthesized the first strand from the qScript^TM^ cDNA Synthesis Kit (Quanta Biosciences, Gaithersburg, MD, USA) using the maximum 8 µL of elute as a starting template. 

We designed optimal primers for termite genes *Relish* and *IMD* and the reference gene *RPL13a* using geniusR8 software (Table 2) [55] and a previously generated termite immune gene sequence from *Z. nevadensis* [54]. We sequenced amplicons generated from each primer pair using an Applied Biosystems 3730XL automated DNA sequencer at the London Regional Genomics Centre (Robarts Research Institute, Western University, Canada) to confirm via sequence homology that the correct target gene was amplified. Finally, we estimated the relative abundance of mRNA between samples using a QX200 Digital Droplet PCR (ddPCR) system (Bio-Rad Laboratories, Inc., Hercules, CA, USA). Briefly, we fractionated individual PCR reaction volumes into ~20,000 nanoliter-sized droplets, then PCR-amplified the fractionated sample using a conventional thermocycler (C100 model, Bio-Rad). The PCR program consisted of an initial five-minute denaturation at 95 °C, followed by 44 cycles of denaturation (95 °C for 30 s) and annealing/extension (53 °C for 60 s). We then stabilized the sample at 4 °C for five minutes, then 90 °C for five minutes with a final hold at 4 °C. Each reaction consisted of 12 μL of EvaGreen Supermix (Bio-Rad, Hercules, CA, USA), 2.5 μL of the forward primer (1 μM), 2.5 μL of the reverse primer (1 μM), and 8 μL of undiluted cDNA template. Following PCR, we estimated the fraction of PCR-positive droplets for each reaction in a flow cytometer. From this information, we used the QuantaSoft (Bio-Rad, Hercules, CA, USA) program to determine the starting DNA template concentration in each sample for all treatments (μg/μL), as well as no-template controls. We replicated the ddPCR experiment five times per treatment (Table 1). 

### 2.4. Quantification of Total Embryonic Protein 

The justification for quantifying embryonic protein was twofold. First, mothers imbue embryos with storage proteins such as vitellogenins (i.e., yolk precursor proteins [56]). We hypothesized that the embryonic protein of immune-elicited (HK-*Sm*) or live-bacteria-injected parents (Live-*Sm*) would be lower relative to that of embryos sired by naïve and saline-treated parents, given expected trade-offs between parental immune activation and offspring provisioning by these parents. Second, the downstream products of the IMD pathway, which responds to Gram-negative bacteria such as *Serratia*, are antimicrobial peptides. Hence, we quantified the embryonic protein content for the following two reasons: to estimate differences in the parental allocation of resources (i.e., embryonic protein) as a function of their treatment, and to control for potential differences in protein content across samples in our antibacterial assay. 

To quantify the total protein concentration of each sample, we used a Bio-Rad Protein Quantification Kit II (cat # 5000002) with bovine serum albumin protein as the standards (ranging from 0 to 0.5 mg/mL in increments of 0.05 mg/mL). For this assay we defrosted and pooled three embryos from the same incipient colony into a single microcentrifuge tube (1.5 mL; Figure 1) containing 214 μL of phosphate-buffered saline (PBS) and 36 μL of the 1X concentration of complete Mini EDTA-free Protease Inhibitor Cocktail (Roche, cat. # 04693159001) to reduce the probability of protein breakdown. We then sonicated (Vibra-Cell^TM^ Model: CU18) each of these pooled samples at an amplitude of 40% for three 5 s pulses to break the chorion and lyse the embryonic cells. We used two 10 μL aliquots as replicates for each homogenate sample. We used the remaining homogenate in the antibacterial assay (Figure 1C; Table 1). 

### 2.5. Antibacterial Assay 

To test if antibacterial activity in embryos is influenced by parental treatment, we incubated embryonic homogenates with the soil bacterium *Arthrobacter* VS10 (Appendix A). *Arthrobacter* is commonly used in eco-immunology studies because it is a relatively slow-growing and more susceptible Gram-negative bacterium relative to other “sturdier” bacteria such as *Serratia* [57,58,59]. Hence, we used *Arthrobacter* instead of *Serratia* as an indicator (i.e., a “*canary in the coal mine*”) to detect evidence of differential embryonic antibacterial activity across parental treatments. We cultured *Arthrobacter* in a manner similar to *S. marcescens*, but all incubations occurred at room temperature. From the stock, we prepared “mini-cultures” by combining 50 μL 1 × 10^8^ CFU/mL live *Arthrobacter* suspended in sterile tryptic soy broth (TSB) with 50 μL of homogenized embryo sample in a well of a 96-well microplate. We then transferred 100 μL of fresh, sterile TSB into each experimental well before incubating the whole 96-well microplate at 25 °C in a BioTek plate reader. We measured the OD_600_ every 30 min for 24 h. We prepared positive and negative controls (Table 3) and calculated the growth rates (GR) from the slopes of the linear portions of the growth curves (OD_600_/minutes).

### 2.6. Statistical Analysis

To calculate differential transcript abundance at each target locus (*Relish* and *IMD*) we first accounted for technical variation in the starting template by normalizing the per-tube concentrations (μg/μL) to that of the stably expressed endogenous reference (*RPL13a*) gene. We then used this normalized standard amount for each gene to estimate expression fold differences as a function of immune treatment, relative to naïve samples. Subsequently, we also performed an overall Kruskal–Wallis test on the normalized standard amount of each gene followed by pairwise comparisons with Mann–Whitney tests, adjusting the significance level with a Bonferroni correction. 

To test the total protein content of two-day-old embryos as a function of queen and king treatments across all three parental effect groups (as fixed effects), we ran a single, general linear mixed-effects model (GLMM). We included the death of the king (fixed), king and queen mass (covariates), total number of embryos produced by day 30 post-pairing (covariate), and king and queen colony of origin (COO; random effects) in the model; these are all factors known to influence the successful establishment of a termite colony [40] (Appendix A). We also tested all possible two-way interactions (Appendix A). COO was used instead of the degree of relatedness (i.e., nestmate vs. non-nestmates) as it allowed us to control for COO-specific variation while also accounting for the genetic relationship between the king and the queen. 

To determine if *Arthrobacter* grown with embryo homogenates exhibited lower growth compared to *Arthrobacter* alone (positive control, *n* = 20), we ran three separate ANOVAs, one for each of the maternal effects, paternal effects, and combined parental effects, with parental treatment as the independent factor. For models that were statistically significant, we applied pairwise Bonferroni corrections (Appendix A). This ANOVA could not test for effects of variables specific to embryo homogenates (e.g., maternal and paternal treatments, king and queen COO, king and queen mass) as these variables were nonexistent in our control treatment (*Arthrobacter* alone). We therefore constructed an additional general linear mixed-effects model (GLMM) to test whether parental treatment differentially affected the antimicrobial activity of embryos. In this model, we were able to account for the effects of total embryonic protein and other factors inherent to the queen or king (Appendix A) on antimicrobial activity. We converted the raw growth rates of our experimental *Arthrobacter* samples into deviations ($GR_{dev}$) from the *Arthorbacter* controls (Appendix A). Using the $GR_{dev}$ values as the dependent variable, we included in the model several fixed factors (queen treatment, king treatment, death or survival of the king), queen COO as a random factor, and several covariates (king and queen mass, day of first egg post-establishment, total protein content). King COO was excluded from this analysis since preliminary models indicated that king and queen COO were collinearly dependent. We retained queen COO, however, as previous analyses of embryonic volume (a proxy measure of embryonic “quality”) revealed that queen COO was the most significant predictor of embryonic volume whereas king COO was insignificant [43]. This model allowed us to assess all three types of transgenerational effects—maternal, paternal, and combined effects—in a single, comprehensive model (Appendix A).

## 3. Results

### 3.1. mRNA Abundance of IMD and Relish

In each case, BLASTX queries of our amplicon sequences confirmed a top hit against the Relish, IMD, and RPL13a genes of the congeneric *Z. nevadensis* genome (E-values < 10^−^^5^, >98% amino acid pairwise identity in each case). The abundance of the endogenous reference gene RPL13a did not vary as a function of treatment (F_2,6_ = 1.83, *p* = 0.24; ANOVA), indicating that it is adequate for normalizing against technical sources of variation. Termite embryos whose mothers were injected with Live-*Sm* showed altered transcript abundance at one of the two immune-related loci, Relish, compared to embryos from naïve and saline-injected (overall Kruskal–Wallis; H = 6.1, df = 2, *p* = 0.046; Figure 2). In pairwise comparisons, the mRNA abundance of Relish in embryos from Live-*Sm* queens was significantly enhanced compared to that in embryos from saline queens (Z = −2.2, df = 1, *p* = 0.03) but not compared to that in embryos from naïve queens (Z = −1.2, df = 1, *p* = 0.2). The abundance of Relish did not differ either between embryos from naïve and saline mothers (Z = 1.0, df = 1, *p* = 0.3). While the IMD gene of embryos from Live-*Sm* queens also exhibited an almost twofold up-regulation relative to naïve and saline mothers, this magnitude of expression was not statistically significant (Kruskal–Wallis; H = 1.5, df = 2, *p* = 0.5; Figure 2).

### 3.2. Total Protein

The total protein concentration within each of the parental treatments was normally distributed (Appendix A). After controlling for the effect of all other variables in the model (Appendix A), the total embryonic protein content did not differ significantly as a function of either king or queen treatment or the combined effects of king and queen treatment (GLMM: F = 0.8, df = 3, 98.5, *p* = 0.5; F = 1.3, df = 3, 99.2, *p* = 0.3; F = 0.2, df = 3, 98.9, *p* = 0.9, respectively). None of the other factors included in the model were significant (Appendix A). Hence, parental contributions toward progeny “quality” (in the form of total protein) do not appear to be influenced by parental pathogenic stress.

### 3.3. Embryonic Antibacterial Properties

The majority of the growth rates were normally distributed (Appendix A), with the exception of Live-*Sm* injected mothers (Shapiro–Wilk = 0.9, df = 20, *p* = 0.05). Overall, two-day-old embryos exhibited antibacterial activity against *Arthrobacter* regardless of parental treatment (maternal effects, F = 11.3, df = 4, 91, *p* < 0.001; paternal effects, F = 8.5, df = 4, 88, *p* < 0.001; combined effects, F = 6.8, df = 4, 91, *p* < 0.001; ANOVA, Figure 3; Appendix A). The antibacterial properties of embryos were consistent across all parental effect experiments (Figure 3; Appendix A).

Contrary to our expectation, parental treatment did not influence embryonic antibacterial activity, even after accounting for other variables in the GLMM model. We detected no significant differences in bacterial growth across maternal or across paternal treatments (F = 0.2, df = 3, 87.8, *p* = 0.9; F = 0.6, df = 3, 88.9, *p* = 0.6, respectively). The antibiotic properties of embryos of the combined maternal and paternal effects experiment did not differ either (F = 1.1, df = 3, 88.4, *p* = 0.4). There were no significant contributions to embryonic antimicrobial activity by other factors and covariates in the model (Appendix A).

## 4. Discussion

Given the paucity of TGIP research in non-model, hemimetabolous insects [5], as well as in social insects generally [32,33,34,35,36,37], we tested for evidence of TGIP in a species of dampwood termite. Termites as a whole have converged in many aspects of their social organization with the social Hymenoptera [60]. Termites are, however, phylogenetically unrelated to that order and are nested within the roach lineage [61]. Therefore, the use of termites may help fill gaps on how TGIP varies as a function of ecological, phylogenetic, and social constraints, and this model should provide an important counterpoint to studies on the social Hymenoptera. 

### 4.1. mRNA Abundance of Relish

Our data indicate that offspring of *Serratia*-infected queens have a significantly greater abundance of *Relish* transcripts compared to embryos of saline (control) queens (Figure 2A). This pattern is consistent with transgenerational immune priming. Similarly, *IMD* tended toward increased abundance in embryos laid by Live-*Sm* queens, but this difference was not statistically significant (Figure 2B), possibly because of minute developmental differences across the five embryos/sample that may have masked precisely timed differences in *IMD* transcript abundance or because *Relish* may have downregulated *IMD* as part of a negative feedback loop. Such negative loops are common in other metabolic pathways [62,63]. Unfortunately, our experimental design does not allow us to distinguish if the transcripts are maternal or embryonic in origin. However, if upregulated transcripts are of maternal origin, then mothers should only need to imbue their egg with one of the two transcripts (i.e., *Relish*) to achieve the same functional results as the *IMD* protein leads to the activation of *Relish* [47]. Finally, instead of provisioning embryos with maternal transcripts, queens may provision their embryos with prefabricated IMD proteins during oogenesis, which, in turn, leads to the production of *Relish* transcripts by the embryo itself [47]. Given that embryos at this early stage of development appear to exhibit only maternally contributed vitellogenins, with no other obvious proteins present [43], we speculate that the *Relish* transcripts are of queen origin. Moreover, embryos from saline-injected queens did not exhibit altered gene expression, indicating that the response is specific to the presence of *Serratia* in the maternal treatment. Regardless of whether transcripts are of maternal or embryonic origin, we showed that the transcription of at least one immune gene (*Relish*) at this early stage of ontogeny is influenced by maternal pathogenic stress and is thus indicative of TGIP.

We elected to compare embryonic differential mRNA abundance only as a function of three out of the four maternal treatments: naïve queens (as a baseline for gene expression), saline queens (to control for the immune-activating effects of cuticular wounding [64,65]), and Live-*Sm* queens. Transcript analyses for embryos laid by HK-*Sm* queens were omitted because insects have been shown to have truncated immune responses when exposed to heat-killed bacteria compared to live bacteria [63]. Furthermore, immune responses following immune elicitation by isolated PAMPs (e.g., lipopolysaccharides, petidoglycans) differ from the responses against full-blown infections with live pathogens [5]. These reasons, along with other constraints, prompted the decision to drop the HK-*Sm* treatment from the transcript analyses. 

### 4.2. Incongruity between Transcriptional and Antibacterial Data

Our transcript analysis showed a significant upregulation of *Relish*. We therefore expected two-day-old embryos laid by Live-*Sm* parents (kings and/or queens) to show enhanced antibacterial activity relative to embryos from naïve and saline-treated parents. Instead, we found that parental treatment did not influence the magnitude of embryonic antibacterial properties. One possibility for this apparent discrepancy between the transcriptional and antibacterial results is that TGIP in termites may be specific to the pathogen, and thus infecting with *Serratia* but testing for inhibition against *Arthrobacter* may have mismatched our assay with our test. There is evidence for immune specificity in insects [66,67], and such specificity may extend to termite TGIP [68]. We tested our embryos against *Arthrobacter* as this genus is a less “robust” Gram-negative bacterium than *Serratia* [57,58]. Thus, *Arthobacter* should have been more sensitive to the embryonic antimicrobial compounds. A second possibility is that testing for protein effectors in two-day-old embryos was too early. TGIP is not necessarily evident in all developmental stages [5]. In *Z. angusticollis*, two-day-old embryos (identified as stage E1 in [43]) laid by naïve queens show only three abundant proteins, likely all maternally contributed vitellogenins [43]. Hence, the abundant embryonic *Relish* transcripts may have not yet been translated into effector proteins at this early stage. Finally, the upregulation of *Relish* may have some role other than immune activation and may thus be independent of the production of antimicrobial compounds. This is unlikely, however, as several studies have directly linked insect *Relish* to the production of antimicrobial peptides [69].

### 4.3. Termite Embryos Have Antimicrobial Properties

We know little about embryonic immunology in termites and the effects of pathogenic stress on their ontogeny [42]. Our data show that termite embryos are capable of inhibiting bacterial growth (Figure 3). It is unclear if the antibacterial properties of two-day-old embryos are exogenous or endogenous to the embryo. Exogenous sources may arise from maternal deposition of antimicrobial compounds on the outer surface of the chorion during the trajectory of embryos through the queen’s reproductive tract [43]. Additionally, once eggs are laid, parents frequently engage in egg-grooming, which is likely accompanied by the deposition of salivary gland secretions known for their antibacterial properties [70,71,72,73]. Embryos may also be immunocompetent through endogenous sources [24,74,75,76,77,78,79], such as vitellogenins, yolk proteins with antibacterial properties [80,81]. Cole et al. [43] reported that *Z. angusticollis* embryos exhibit constitutive antifungal properties. Matsuura et al. [72] reported that eggs of the subterranean termite *Reticulitermes* contained lysozymes. Although lysozymes are known to have antimicrobial properties [82,83], the Matsuura et al. study never tested these lysozymes against bacteria. Taken together, the current transcript and embryonic antibacterial results, combined with data of intra-embryonic antifungal properties [43], indicate that termite embryos possess endogenous protection against pathogens. The existence of broad protection against both bacteria and fungi lends support to the view that pathogens pose strong selection pressures to all members of a termite colony, including embryos, the most immature of stages [43]. More work is needed to identify the protein(s) responsible for embryonic antibacterial and antifungal activity.

Total protein did not correlate with antibacterial activity, suggesting that future work investigating TGIP in termites should investigate specific proteins (Appendix A). Total protein also did not differ as a function of parental treatment (Appendix A). These results are consistent with previous work showing that embryo volume did not differ across parental treatments [42]. Embryonic volume and protein content—two measures of resource investment into egg quality by insect parents—appear to be resilient to the effects of parental pathogenic stress, supporting the conclusion that infected and resource-limited queens prioritize egg “quality” over quantity when forced to make physiological decisions between their reproduction and their own immune activation (i.e., trade-offs) [42].

## 5. Conclusions

We provide evidence showing that in the face of disease, termite queens increase the abundance of *Relish* transcripts in their embryos, a pattern consistent with TGIP. The second immune-related gene, *IMD*, was also upregulated, although these differences were not statistically significant. These results suggest that infected queens can influence the immune protection of their first embryos by transferring either immune-related proteins, transcripts, or epigenetic markers [54,84]. Contrary to expectation, the differential immune gene transcript abundance as a function of maternal treatment did not correspond to higher antibiotic properties in embryos, pointing to translation of functional gene products occurring later in embryonic development or perhaps indicating that the antibiotic effector is masked by the high antibiotic properties of other compounds present in two-day-old embryos (i.e., maternally contributed vitellogenins). Work is underway to further test for evidence of TGIP at the protein level and in later developmental stages of *Z. angusticollis* from both heat-killed and live pathogenic exposures.

## Figures and Tables

**Figure 1 insects-11-00149-f001:**
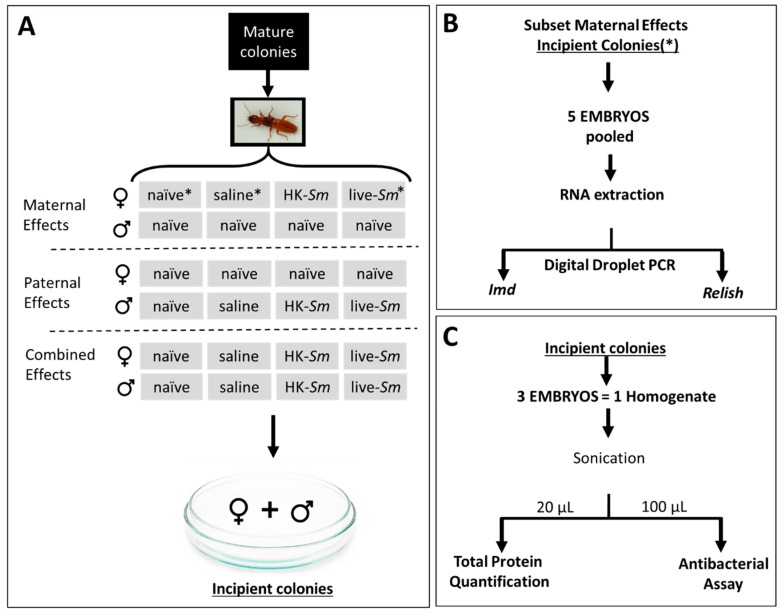
Experimental scheme and protocols. (**A**) Virgin alates were collected from mature colonies, de-winged, and weighed. Parental effects (either maternal, paternal, or combined maternal and paternal) following *Serratia* exposure were tested by treating de-winged reproductives with one of three injections—saline, heat-killed *Serratia* (HK-*Sm*), or live *Serratia* (live-*Sm*)—or leaving the reproductives untreated (naïve). These individuals were then paired with mates to form incipient colonies. Pairs were set up in Petri dishes with 2.5 mg of white birch wood and dampened Whatman #1 filter paper. (**B**) In a subset of the maternally treated colonies (where naïve, saline, and Live-*Sm* queens were paired with naïve kings, indicated with the symbol * in **A**), five embryos (two days old) per colony were pooled into a single sample and subsequently used in RNA extraction and Digital Droplet PCR to quantify two immune-related genes. (**C**) Following the onset of oviposition, two-day-old embryos resulting from all pairings in (**A**) were collected. Three embryos per incipient colony were pooled into a single homogenized sample. This pooled sample was divided in two aliquots, one to quantify protein content and the second to test that sample’s antibacterial properties. Sample sizes for (**A–C**) are given in Table 1.

**Figure 2 insects-11-00149-f002:**
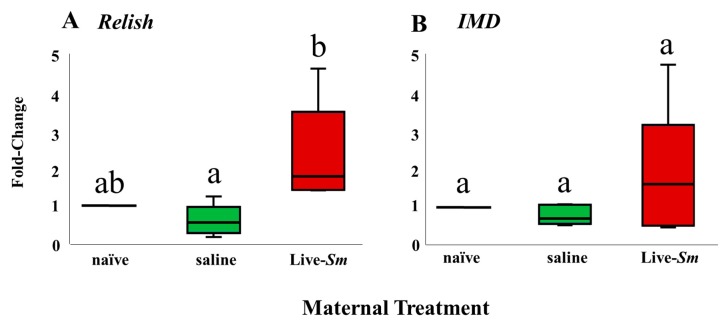
Fold transcript abundance differences of two immune-related loci, *Relish* (**a**) and *IMD* (**b**), as a function of maternal treatment; *n* = 5 samples consisting of five pooled embryos. Bars (±SEM) represent gene dysregulation relative to “naïve” samples (which were normalized to 1). Different letters denote statistical significance across pairwise comparisons for all three treatments.

**Figure 3 insects-11-00149-f003:**
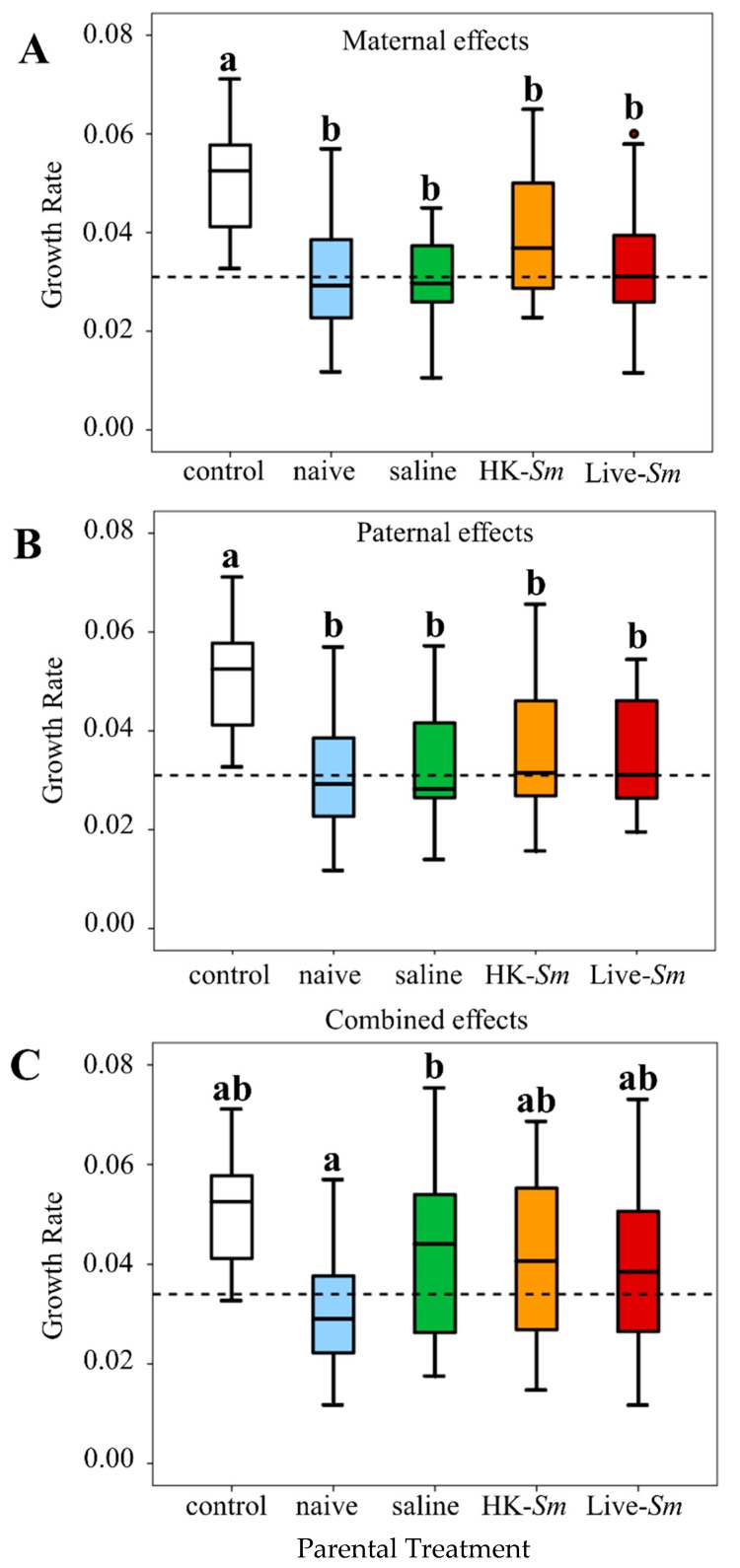
Median growth rates (GR) of *Arthrobacter* cultured with and without embryonic homogenates from (**A**) maternal effects, (**B**) paternal Effects, and (**C**) combined effects. Letters indicate statistical significance (Appendix A). Dashed lines indicate the median growth rate of *Arthrobacter* grown in the presence of embryo homogenates across the four parental treatments for each graph.

**Table 1 insects-11-00149-t001:** Sample sizes by parental treatment.

	Parental Treatment ^1^	Incipient Colonies Established	Samples for Digital Droplet PCR ^2^	Samples for Total Protein ^3^	Samples for Antibacterial Assay ^3^
	NQ + NK	66	5	33	33
	SQ + NK	45	5	10	10
**Maternal Effects**	HQ + NK	45	--	15	15
	LQ + NK	97	5	17	17
	NQ + SK	41	--	11	11
**Paternal Effects**	NQ + HK	43	--	15	15
	NQ + LK	78	--	11	11
	SQ + SK	58	--	21	21
**Combined Effects**	HQ + HK	51	--	12	12
	LQ + LK	99	--	11	11
	Total	623	15	156	156

^1^ N = naïve, S = saline, H = HK-*Sm*, L = Live-*Sm*, Q = queen, K = king. ^2^ One sample comprised five embryos pooled from the same incipient colony. ^3^ One sample included three embryos pooled from the same incipient colony. Of the queens that survived the treatment, only 68% yielded eggs [42]. While we were able to collect 1138 eggs in total [42], incipient colonies with fewer than 3 eggs were excluded due to the sensitivity limitations of our assays.

**Table 2 insects-11-00149-t002:** Primer sequences used to assess immune gene transcription.

Locus (Accession)		Primer Sequence (5′–3′)	Amplicon (bp)
*Relish* (Znev_11193)	F	TCT GCA CAC TCC TGC TTA AA	137
	R	AAT CAT CAT CAC TCT CCG GC	
*IMD* (Znev_02405)	F	GTG CAA AAT TCT CCC AGT ACA	148
	R	CTC TCC AAT GTT CTC CGA CA	
*RPL13a* (Znev_00068)	F	CAC TTC AGA GCA CCA AGC AA	152
	R	ACG TTT CAA TGC TGC CTT TC	

**Table 3 insects-11-00149-t003:** Controls for the antimicrobial assay with embryo samples.

Control	Well Contents
+ control for bacterial growth	150 μL TSB + 50 μL of *Arthrobacter* suspension
- control	200 μL TSB
- control	50 μL embryo sonicate + 150 μL sterile water
- control	200 μL sterile water

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
