# Peer review of "Relish as a Candidate Marker for Transgenerational Immune Priming in a Dampwood Termite (Blattodae: Archeotermopsidae)"

_insects, 2020, doi:10.3390/insects11030149_

Round 1

Reviewer 1 Report

I am happy with authors' explanations, which properly addresses my concerns, and also their actions on revising the manuscript - therefore I recommend to accept this manuscript as it is. Some minor spelling checks may be required.

Author Response

We thank Reviewer 1 for their feedback. We have carefully gone through the ms and corrected any remaining errors we could find. All corrections are highlighted in yellow in the revised ms.

-The authors.

Reviewer 2 Report

The authors have satisfactorily responded to all my concerns. 

Typos in author names in ref 37 should be corrected.  

Author Response

We thank Reviewer 2 for their feedback. We have carefully gone through the ms and corrected any errors we could find, including reference 37. All corrections are highlighted in yellow in the revised ms.

-The authors.

Reviewer 3 Report

This is an interesting and relevant paper. The methods are sound and the results are well discussed. I don't have any recommendations at this point and this would be suitable for this journal.  

Author Response

We thank Reviewer 3 for their feedback. We have carefully gone through the ms and corrected any minor errors we could find. All corrections are highlighted in yellow in the revised ms.

-The authors.

Reviewer 4 Report

   In the work of Cole et al. entitled “Relish as a candidate marker for transgenerational immune priming in a dampwood termite (Blattodae: Archeotermopsidae)”, the authors have assessed the role of two genes of the innate immune response in transgenerational immune priming (TGIP). For that purpose, the offspring of Serratia-injected termites was assayed for gene expression of IMD and Relish, antibacterial activity and protein expression. The manuscript is well written and the Reviewers 1 to 3 have covered all my major concerns. Even though the authors rebutted every point raised, no additional experiments were conducted under the (understandable) rationale of financial constraints and insect availability. That gives little room for experimental improvement and further suggestions of assays. As stated by Reviewer 3, I am also puzzled by the choice of not testing antibacterial activity against the same pathogen, even though the authors explained that the pathogen used was more sensitive.

Specific comments

- In the abstract, the authors express that “Taken together, Z. angusticollis shows transcriptional but not translational evidence for TGIP”. With the methodology employed (determination of total proteins) it is very difficult to conclude that there is no translational evidence of TGIP because it is very possible that the expression of some specific proteins was indeed altered. The authors should tone down this conclusion to match the scope of their findings. I suggest reword or delete the part of “translational”.

- The authors use the word “transcriptomic” throughout the manuscript meaning transcriptional. Transcriptomics takes into account the sum of all RNA transcripts of an organism and not just three genes. Please correct.

- Page 6, line 148: The scientific name of Z. nevadensis is misspelled. Please correct.

- Page 7, lines 164-167: The sequences of the genes amplified should be available, i.e. annotated in a data bank or at least provided as a supplementary material. What was the percentage of identity of the Z. angusticollis sequences when compared with that of Z. nevadensis? This data is relevant because it gives us an idea of how well these proteins are conserved.

- Page 7, lines 181-187: Why did the authors use quantification of total proteins and not study the expression of specific proteins? The latter approach could have been performed using western blot and polyclonal antibodies against vitellogenin or antimicrobial peptides. Vitellogenin is well conserved and antibodies raised against the protein of a different species could be employed.

- Page 15, lines 378-380: it is hard to grasp the meaning of that sentence without going through to the literature cited. Please, elaborate on that point and develop the idea.

- Since the authors have the cDNA, it would be a relevant addition to the manuscript to test other genes, for example antimicrobial peptides.

- I do not understand the decision of not pursuing the data of heat-killed pathogens on the grounds of “no consistent results”. Was the error bar too big? That could just mean biological variation. This should be better explained in the discussion section.

Author Response

Reviewer 4 in Black

Author Responses in Purple

In the work of Cole et al. entitled “Relish as a candidate marker for transgenerational immune priming in a dampwood termite (Blattodae: Archeotermopsidae)”, the authors have assessed the role of two genes of the innate immune response in transgenerational immune priming (TGIP). For that purpose, the offspring of Serratia-injected termites was assayed for gene expression of IMD and Relish, antibacterial activity and protein expression. The manuscript is well written and the Reviewers 1 to 3 have covered all my major concerns. Even though the authors rebutted every point raised, no additional experiments were conducted under the (understandable) rationale of financial constraints and insect availability. That gives little room for experimental improvement and further suggestions of assays. As stated by Reviewer 3, I am also puzzled by the choice of not testing antibacterial activity against the same pathogen, even though the authors explained that the pathogen used was more sensitive.

We thank Reviewer 4 for their time and feedback. We have taken all of the reviewers’ feedback into consideration in designing our next generation of experiments that build of current findings. In the ongoing experiments, we have considered how pathogen-specific immune responses might affect our assay and thus, affect our ability to detect pathogen-specific immune priming.

Specific comments

- In the abstract, the authors express that “Taken together, Z. angusticollis shows transcriptional but not translational evidence for TGIP”. With the methodology employed (determination of total proteins) it is very difficult to conclude that there is no translational evidence of TGIP because it is very possible that the expression of some specific proteins was indeed altered. The authors should tone down this conclusion to match the scope of their findings. I suggest reword or delete the part of “translational”.

We agree that some specific proteins might have been altered but we cannot claim to have observed this response. Therefore, for both Relsih and IMD,  Our conclusion that we show ' transcriptional but not translational evidence for TGIP' is correct. To clarify, we have altered the last sentence of the abstract to read:

“In total we provide the first suggestive evidence of transgenerational immune priming in a termite.”

- The authors use the word “transcriptomic” throughout the manuscript meaning transcriptional. Transcriptomics takes into account the sum of all RNA transcripts of an organism and not just three genes. Please correct.

We have correct this error throughout the ms.

- Page 6, line 148: The scientific name of Z. nevadensis is misspelled. Please correct.

Thank you for catching this error, it has been fixed.

- Page 7, lines 164-167: The sequences of the genes amplified should be available, i.e. annotated in a data bank or at least provided as a supplementary material. What was the percentage of identity of the Z. angusticollis sequences when compared with that of Z. nevadensis? This data is relevant because it gives us an idea of how well these proteins are conserved.

Only the Z. angusticollis mitochondrial genome has been sequenced. Hence, we cannot make the suggested comparisons to Z. nevadensis. We can only make inferences about Z. angusticollis’ immune function based on the one publication of the genome of Z. nevadensis.

- Page 7, lines 181-187: Why did the authors use quantification of total proteins and not study the expression of specific proteins? The latter approach could have been performed using western blot and polyclonal antibodies against vitellogenin or antimicrobial peptides. Vitellogenin is well conserved and antibodies raised against the protein of a different species could be employed.

The quantification of total protein was required to statistically control for differences in protein content across all our samples for the antibacterial assay. Our methods struck a balance between sample size vs. concentration vs. number of assays run on a single sample. Basically, we could not dilute the samples into a larger volume of saline because the assays were not sensitive to detect such minute amounts of protein. As it is, we were forced to pool  three embryos per sample. This was unfortunate because studies suggest that insects do not provision all of their eggs equally (e.g., Rosengaus et al., 2017). As a result, we had a limited volume for each sample to work with. Future studies will take into consideration protein expression. We have altered our final line (397-399) to read:

“Work is underway to obtain evidence of functional TGIP in later developmental stages of Z. angusticollis from both heat-killed and live pathogenic exposures, and to identify the mechanism and degree of specificity underlying this cross-generational response.”

- Page 15, lines 378-380: it is hard to grasp the meaning of that sentence without going through to the literature cited. Please, elaborate on that point and develop the idea.

We have revised this paragraph to clarify. It now reads:

“Total protein did not correlate with antibacterial activity, suggesting that future work investigating TGIP in termites should investigate specific proteins (Supplemental Table 5). Total protein also did not differ as a function of parental treatment (Supplemental Table 2). These results are consistent with previous work showing that embryo volume did not differ across parental treatments [42]. Embryonic volume and protein content –two measures of resource investment into egg quality by insect parents—appear to be resilient to the effects of parental pathogenic stress, supporting the conclusion that infected and resource-limited queens prioritize egg “quality” over quantity when forced to make physiological decisions between their reproduction and their own immune activation (i.e., trade-offs) [42].”

- Since the authors have the cDNA, it would be a relevant addition to the manuscript to test other genes, for example antimicrobial peptides.

It would be a great idea to screen a larger number of genes. We have plans to do so in future, with new students and new funding.

- I do not understand the decision of not pursuing the data of heat-killed pathogens on the grounds of “no consistent results”. Was the error bar too big? That could just mean biological variation. This should be better explained in the discussion section.

The heat-killed pathogen treated samples did not amplify properly despite much effort to optimize the PCR. The amount of template from embryos was very limited and unfortunately, we had to drop this treatment.

Reviewer 5 Report

The authors studied for the first time trans-generational immune priming in a hemimetabolous insect. The subject is worthy to be investigated and of interest, but the experimental design is relatively poor. One weak point of the study is the low number of immunity-related markers tested. Why was not the expression of immunity-related effector molecules such as genes encoding antimicrobial peptides, lysozyme or phenoloxidase analyzed? Another weak point is that only a single bacterial species has been used to determine antibacterial activity. Why was not a Gram-positive bacterium such as Micrococcus luteus used? The latter indicate very sensitive e.g. lysozyme activity. Taken together, the authors show only that embryos from mothers injected with bacteria display a moderately enhanced level of relish expression when compared with controls. This minor difference in expression of an immunity-related signaling molecule was not translated in higher antibacterial activity in embryos. Thus, the conclusion that TGIP occurs in termites is not really supported by the data. The manuscript is well-written, but the outcome is rather poor. Lane 143: replace effector by signaling molecules as effector molecules are widely used for antimicrobial peptides.

Author Response

We thank Reviewer 5 for their time and feedback. We agree it would be a great idea to screen a larger number of genes. We have plans to do so in future, with new students and new funding. Our goal here was keep things tightly focused and give proof of principle for future work. Our results are modest, but scientifically sound and interesting.

Regarding the second weak point: Incipient termite colonies are challenging to work with, and limits the scope of individual experiments. Here is our response to Reviewer 3 during the first round of reviews, which addresses the same issue in detail:

“Incipient colonies of Zootermopsis angusticollis only have a 40-60% survivorship through the first 30 days post-establishment in the lab (Cole et al., 2018; Cole and Rosengaus, 2019; Calleri et al., 2005, 2006, 2007). Of the surviving colonies, only a small percentage yield eggs (Cole et al., 2018; Cole and Rosengaus, 2019). Typically, incipient colonies will produce between 1 and 30 eggs in the first brood. Since our assay required a minimum of 3 eggs, this further reduced the number of incipient colonies from which we collected samples. Since each sample was aliquoted to different assays, the sample could only be used against a single pathogen in the antibacterial assay. In the present study, we set up 623 colonies which were followed for 30-days in order to collect all of eggs produced. Perhaps if we had duplicated the number of incipient colonies, we could have secondary tests against multiple pathogens to test for the degree of specificity of TGIP. Instead, we opted to test for maternal and paternal effects against the same pathogen. We agree that functionally testing the degree of specificity in termite TGIP would be a worthwhile avenue of future research.”